# DESS deconstructed: Is EDTA solely responsible for protection of high molecular weight DNA in this common tissue preservative?

Amy Sharpe[1]☯, Sonia Barrios[1], Sarah Gayer[1], Elisha Allan-Perkins[2], David Stein[1], Hannah J. Appiah-Madson[1]☯, Rosalia Falco[1]☯, Daniel L. Distel[1]*

1 Ocean Genome Legacy Center, Northeastern University, Nahant, Massachusetts, United States of America, 2 Brookline, New Hampshire, United States of America

☯ These authors contributed equally to this work.
* d.distel@northeastern.edu

**Data Availability Statement:** All specimen data are available from the Ocean Genome Legacy Center database (accession numbers S29192 – S29215

## Abstract

DESS is a formulation widely used to preserve DNA in biological tissue samples. Although it contains three ingredients, dimethyl sulfoxide (DMSO), ethylenediaminetetraacetic acid (EDTA) and sodium chloride (NaCl), it is frequently referred to as a DMSO-based preservative. The effectiveness of DESS has been confirmed for a variety of taxa and tissues, however, to our knowledge, the contributions of each component of DESS to DNA preservation have not been evaluated. To address this question, we stored tissues of three aquatic taxa, *Mytilus edulis* (blue mussel), *Faxonius virilis* (virile crayfish) and *Alitta virens* (clam worm) in DESS, each component of DESS individually and solutions containing all combinations of two components of DESS. After storage at room temperature for intervals ranging from one day to six months, we extracted DNA from each tissue and measured the percentage of high molecular weight (HMW) DNA recovered (%R) and normalized HMW DNA yield (nY). Here, HMW DNA is defined as fragments >10 kb. For comparison, we also measured the % R and nY of HMW DNA from extracts of fresh tissues and those stored in 95% EtOH over the same time intervals. We found that in cases where DESS performed most effectively (yielding ≥ 20%R of HMW DNA), all solutions containing EDTA were as or more effective than DESS. Conversely, in cases where DESS performed more poorly, none of the six DESS-variant storage solutions provided better protection of HMW DNA than DESS. Moreover, for all taxa and storage intervals longer than one day, tissues stored in solutions containing DMSO alone, NaCl alone or DMSO and NaCl in combination resulted in %R and nY of HMW DNA significantly lower than those of fresh tissues. These results indicate that for the taxa, solutions and time intervals examined, only EDTA contributed directly to preservation of high molecular weight DNA.

and S29350 – S29455). DNA sequences were deposited in the Barcode of Life Datasystem (BOLD) under accession numbers DESS001-19 through DESS006-19 and DESS007-20 through DESS032-20. All other relevant data are within the paper and its Supporting Information files.

**Funding:** This work was funded by the Richard Lounsbery Foundation (DLD), Francis Goelet Charitable Lead Trusts (DLD), Ocean Genome Legacy Center Operations Fund (DLD), NSF FMSL program (DBI 1722553, to Northeastern University). The funders had no role in study design, data collection and analysis, decision to publish or preparation of the manuscript.

**Competing interests:** The authors have declared that no competing interests exist.

## Introduction

The growing importance of DNA-based research has created increasing demand for methods that can preserve high-quality DNA in biological samples. A number of available preservation techniques can delay the degradation of DNA in tissue samples (reviewed in[1, 2]) but many are not easily adapted to the wide range of conditions commonly encountered by researchers working in the field. For example, cryopreservation is considered to be among the best techniques for preserving DNA in tissue,[2–5] but mechanical freezers and freezing agents such as dry ice and liquid nitrogen are expensive, bulky, hazardous and often subject to transportation and shipping restrictions. Similarly, ethanol (EtOH) is one of the most commonly used preservatives,[1, 6, 7] but is flammable, toxic, considered a controlled substance in many jurisdictions and may work best at cold temperatures.[8, 9] EtOH is also frequently subject to legal and travel restrictions.[10, 11] While a variety of commercial products are available for DNA preservation, these formulations are typically proprietary, expensive, not amenable to user modification and incompletely documented in peer-reviewed literature. Hence, practical and well-documented solutions for field preservation of DNA in tissues are in high demand.

In 1991, Seutin, White[12] introduced a liquid preservative solution that has become widely known as DMSO-salt or DESS. The acronym DESS reflects the composition of this formulation, an aqueous solution containing 20% dimethyl sulfoxide (DMSO), 0.25M ethylenediaminetetraacetic acid (EDTA) and saturated sodium chloride (NaCl), adjusted to pH 8.0. Although supporting data were not provided, the authors proposed that EDTA and NaCl may contribute to DNA preservation in tissue by chelating divalent cations required for the activity of nucleases and by denaturing nuclease enzymes, respectively, while DMSO may serve as a penetrant, helping to facilitate transport of these ingredients into cells.[12]

Several qualities make DESS a desirable preservative for field applications. At the concentrations used, the components of DESS have low toxicity and present low risks of fire and explosion. Additionally, DESS is simple and inexpensive to prepare and can be stored and used at room temperature. Tissues stored in DESS have been reported to yield DNA of a similar quality and quantity as tissues preserved cryogenically or in other chemical preservatives. [4, 7, 12] Moreover, researchers have routinely conducted a variety of common analyses including spectrophotometric analysis,[13] Southern blots,[12] PCR amplifications,[4, 7, 9, 10, 13–17] fragment analysis,[9, 17] qPCR,[7, 13, 18] Sanger sequencing[15] and Illumina sequencing[19] using DNA extracted from tissues stored in DESS. Furthermore, DESS has been tested on a variety of organisms including jellyfish, anemones, snails and worms,[4] nematodes,[15, 17] corals,[7] coral microbial communities,[9, 19] fish,[10] cetaceans,[14] bats,[13] birds,[12] mice,[11] humans,[18] pigs[16] and fecal samples from baboons;[20] in most cases comparing favorably to other tested preservation methods over time intervals ranging from less than 1 day to more than 15 years. In our review of publications assessing the effectiveness of DESS, the median preservation period was 6 months, indicating that DESS can preserve DNA over time intervals suitable for many research applications.[4, 7, 9–12, 14–18, 20] As a result, the original recipe of Seutin, White[12] has come into wide use, largely without modification.

Despite the success of DESS in many field applications, there are some concerns associated with its use. Although DMSO has low toxicity (LD50 in rat = 14,500 mg/kg, oral; 40,000 mg/kg, skin), it readily penetrates skin and may enhance the absorption of many potentially harmful chemicals into the bloodstream through skin contact. Thus, DMSO may become a serious health hazard if inadvertently combined with toxic materials.[21] Concentrated DMSO is also flammable and an irritant, adding risk to the preparation of DESS.[1] EDTA and NaCl also have low toxicity (oral LD50 in rat = 2,000 mg/kg and 3,000 mg/kg, respectively). Both are

used widely as additives in food, drugs and cosmetics, are nonflammable, chemically stable and are not known carcinogens. Nonetheless, gloves and eye protection are recommended when using DESS and its components. Because it is a saturated NaCl solution, DESS may be prone to precipitation, which may hamper the recovery of small or delicate samples. Finally, DMSO freezes at just below room temperature (19°C) potentially limiting the usefulness of DESS at cold temperatures.

Although DESS is often referred to in literature as a DMSO-based preservative, e.g. DMSO salt-saturated solution,[22] salt-saturated DMSO,[7, 13] DMSO-salt[4, 10–12] or simply DMSO,[2, 13, 14, 16, 18, 19] to our knowledge the contributions of the individual components of DESS to DNA preservation have not been examined systematically. Here we examine the extent to which each of the ingredients of DESS, individually and in combination, contribute to the preservation of DNA in the tissues of three common aquatic organisms, *Mytilus edulis* (blue mussel), *Faxonius virilis* (virile crayfish) and *Alitta virens* (clam worm), under typical field and laboratory conditions. To our knowledge, this is the first report to evaluate the ingredients of this popular DNA preservative formulation using a factorial experimental design.

## Methods

### Ethics statement

This study was carried out in accordance with the recommendations in the Guide for the Care and Use of Laboratory Animals of the National Institutes of Health and Northeastern University's Institutional Animal Care and Use Committee policies. Although no vertebrate animals were used in this study, the general principles of humane animal care were applied to the invertebrate animals used.

### Experimental design

To evaluate the contributions of the three components of DESS to DNA preservation in tissues, DNA quality was compared among extracts of tissues from three aquatic invertebrate taxa preserved for various time intervals in DESS or one of six DESS-variant solutions in a full factorial design. Each of these six variants contained either one of the three components of DESS individually or one of the three possible pairwise combinations of two DESS components. When preparing these solutions, DMSO and EDTA were maintained in the same concentrations as they appear in DESS and, when present, NaCl was added to saturation (Table 1). All solutions were prepared at room temperature (22–24°C) using distilled deionized water as the diluent. Hereafter, we use the following abbreviations to indicate the components of each storage solution: 20% DMSO (D), 0.25M EDTA (E) and saturated NaCl (SS; Table 1). In addition, extracts from each of the treatments were compared to those obtained from tissues preserved in 95% EtOH and fresh tissues extracted immediately after specimens were euthanized.

### Taxon selection and sourcing

The three taxa, *Mytilus edulis* Linnaeus (blue mussel; *N* = 35), *Faxonius virilis* Hagan (virile crayfish; *N* = 35) and *Alitta virens* M. Sars (clam worm; *N* = 50) selected for this study represent the common aquatic invertebrate phyla Mollusca, Arthropoda and Annelida, respectively. Previous work by Dawson, Raskoff[4] and our own observations indicated that DESS preserved DNA well in a variety of similar mollusks and arthropods but performed poorly on nereid worms closely related to *A. virens*. Therefore, the selected taxa likely represent both good and poor use cases for preservation in DESS. *Mytilus edulis* were collected at the Seaport Landing Marina in Lynn, MA (42.45859 N, -70.94275 W) under the Commonwealth of

**Table 1. Tissue storage solutions used in this study.**

|  | Ingredients* | | | |
| --- | --- | --- | --- | --- |
|  | **DMSO (ml)** | **0.5M EDTA (ml)** | **NaCl (g)$^\alpha$** | **EtOH (%)** |
| *DESS* | 100 | 250 | 105 | - |
| *DE* | 100 | 250 | - | - |
| *DSS* | 100 | - | 125 | - |
| *ESS* | - | 250 | 155 | - |
| *D* | 100 | - | - | - |
| *E* | - | 250 | - | - |
| *SS* | - | - | 180 | - |
| *EtOH* | - | - | - | 95 |

DMSO (D), dimethyl sulfoxide; EDTA (E), ethylenediaminetetraacetic acid; NaCl (SS), sodium chloride; EtOH, ethanol.

*Stocks were diluted to 500 ml final volume with distilled deionized water.

$^\alpha$ Approximate quantities required to reach saturation.

Massachusetts Division of Marine Fisheries Scientific Collection Permit #156386. *Faxonius virilis* were purchased live from A. J.'s Bait & Tackle in Meredith, NH. Live *A. virens* were purchased from Al's Bait and Tackle in Beverly, MA. Samples of all specimens were deposited into the Ocean Genome Legacy Center (OGL) collection and can be accessed using specimen IDs S29192-S29215 and S29350-S29455.[23]

The taxonomic identities of the specimens used in this study were determined by traditional morphology-based methods and confirmed by "DNA barcode" analysis.[24] The *COI* barcoding region as identified by Hebert, Ratnasingham[24] was amplified from DNA extracts of two specimens per taxa using LCO1490_t1 (5′-TGTAAAACGACGGCCAGTGGTCAACAAA TCATAAAGATATTGG- 3′) and HCO2198_t2 (5′-CAGGAAACAGCTATGACTAAACTTCA GGGTGACCAAAAAATCA- 3′) primers.[25] Each PCR amplification contained 2 μl DNA template, 17.5 μl One*Taq* 2X Master Mix (New England BioLabs; Ipswich, MA), 10 μM of both forward and reverse primers and was brought to 35 μl total volume with deionized water. PCR thermocycler conditions were initiated with a heated lid at 94°C for 30 seconds, followed by 30 cycles of 94°C for 30 seconds, 52°C for 40 seconds and 68°C for 60 seconds, with a final extension at 68°C for 5 minutes using a PCT-200 thermocycler (MJ Research, Inc.; Waltham, MA). PCR success was visualized by 1% agarose gel electrophoresis (see details below) and 10 μl of each amplicon was bi-directionally sequenced by the Sanger method on an Applied Biosystems 3730xl DNA Analyzer (Foster City, CA) at a commercial sequencing facility (GEN-EWIZ, South Plainfield, NJ). Resulting sequences were edited and analyzed using Geneious v.8 (Auckland, New Zealand), automatically trimming ends to remove sequence with greater than a 1% chance of error per base and setting 500 bp as a minimum threshold for a successful read. Assembled contigs were deposited in the Barcode of Life Datasystem (BOLD) under accession numbers DESS001-19 through DESS006-19. Sequence identities to best matches in BOLD are reported in S1 Table.

## Dissection and tissue sampling

Live specimens of *M. edulis* and *F. virilis* were stored on ice and live *A. virens* were stored at 4°C prior to dissection. Gill, abdominal muscle and body tissue samples were collected immediately after euthanasia from specimens of *M. edulis*, *F. virilis* and *A. virens*, respectively. Nine tissue subsamples of approximately 100 mg (avg. 91.94 ± 24.47 mg) each were collected from

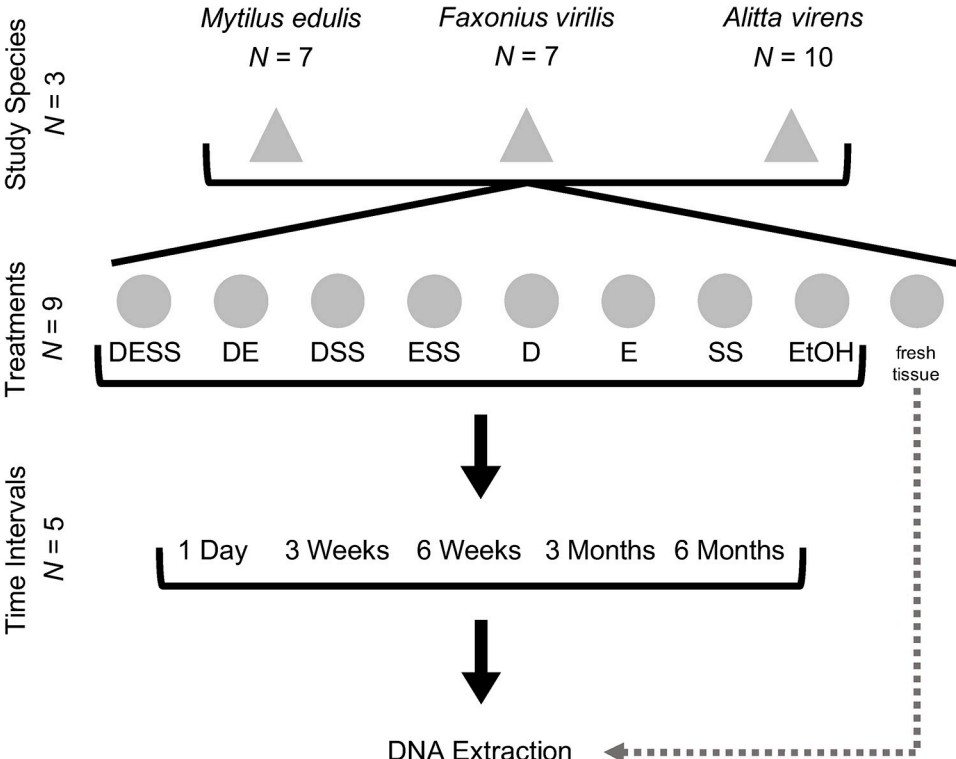

**Fig 1. Experimental design.** Seven specimens of *Mytilus edulis*, seven specimens of *Faxonius virilis* and ten specimens of *Alitta virens* were sampled for each time interval. Nine tissue subsamples were collected from each specimen. DNA was extracted from one subsample immediately after dissection without preservation (fresh tissue). Of the remaining eight, one was stored in DESS, one in each of six DESS-variant solutions (DE, DSS, ESS, D, E and SS) and one in 95% EtOH. Separate specimens of each taxon were used for each time interval, for a total of 35 *M. edulis*, 35 *F. virilis* and 50 *A. virens*.

each specimen. Eight of the subsamples from each specimen were distributed into 1.8 mL cryotubes each containing either 1 mL of DESS, one of the six DESS-variant solutions or 95% EtOH and were stored at room temperature. DNA was extracted from the ninth subsample, hereafter referred to as fresh tissue, immediately after dissection without preservation. To ensure selection of a time course within which samples could be assessed before reaching an unquantifiable state of degradation, storage intervals of 1 day, 3 weeks, 6 weeks, 3 months and 6 months were chosen (Fig 1). Gel electrophoresis indicated progressively increasing but qualitatively similar trends of degradation among taxa and treatments with increasing time and showed that HMW DNA could be detected for at least one treatment for all taxa and time intervals examined (S1 Fig). Therefore, the first, middle and last time intervals, 1 day, 3 months and 6 months, were chosen for further qualitative and quantitative analyses. Seven specimens of *M. edulis*, seven specimens of *F. virilis* and ten specimens of *A. virens* were assigned to each time interval for a total of 35 *M. edulis*, 35 *F. virilis* and 50 *A. virens*. Across all treatments, taxa and time intervals, a total of 1,080 samples were analyzed, including 315 samples from *M. edulis*, 315 samples from *F. virilis* and 450 samples from *A. virens*.

## DNA extraction and quantification

After the assigned storage interval, each tissue sample was removed from its storage solution and reweighed. A tissue subsample of approximately 30 mg (avg. 29.16 ± 4.34 mg) was then

excised for DNA extraction. To control for changes in tissue weight during storage, a correction ratio was calculated by dividing the tissue weight prior to storage by the post storage weight. Each extract subsample weight was then multiplied by the correction ratio to estimate the fresh tissue weight equivalent for that subsample. These values were then used to calculate normalized yield (see statistical analysis). DNA was extracted from tissues using the DNeasy Blood and Tissue Kit (Qiagen; Hilden, Germany) following the manufacturers recommended protocol with tissues digested overnight and DNA eluted by adding two sequential 50 μl volumes of Buffer AE for a total elution volume of 100 μl. DNA was extracted from the ninth fresh tissue subsample immediately after dissection by excising and weighing tissue and placing it directly into the digestion solution.

In this investigation, high molecular weight (HMW) DNA is defined as DNA fragments greater than 10 kb in length. The presence of HMW DNA in each extract was determined qualitatively by agarose gel electrophoresis and quantitatively using an Agilent Technologies TapeStation 2200 DNA Analyzer (Santa Clara, CA). For agarose gel electrophoresis, 3 μl (avg. 0.27 ± 0.70 μg; S2 Table) of each DNA extract was loaded on a 20 cm horizontal slab gel (1% agarose, 1x TAE buffer containing 1% GelRed nucleic acid gel stain (Biotium; Fremont, CA)) and separated at approximately 3 v/cm for 60 minutes and then visualized using a BioRad Gel Doc XR+ Molecular Imager and Image Lab software (Hercules, CA). The first and last lane of each gel was loaded with 1.5 μl of λ DNA-HindIII Digest DNA Ladder (500 μg/ml; New England BioLabs; Ipswich, MA) as a molecular weight standard. Quantitative analyses were performed on 1 μl (avg. 0.09 ± 0.23 μg; S2 Table) of DNA extracts from the 1 day, 3 month and 6 month time intervals using the Agilent DNA Analyzer genomic DNA ScreenTapes and TapeStation Analysis Software (V.A.02.02 (SR1)) to determine both the percent recovery (%R) and normalized yield (nY) of HMW DNA. Additionally, 2 μl (avg. 0.18 ± 0.46 μg; S2 Table) of each sample were analyzed using a Nanodrop 1000 droplet spectrophotometer (Thermo Fisher Scientific; Waltham, MA) to estimate DNA purity using the absorbance ratio at A260/A280.

To evaluate their equivalence with respect to PCR amplification and sequencing, the barcode region of the mitochondrial *COI* gene was amplified and sequenced from three randomly selected DNA extracts from fresh tissues and tissues stored for 6 months in DESS and E, as described above.

## Statistical analysis

The percentage of HMW DNA recovered (%R) was calculated using data obtained from the Agilent Technologies TapeStation 2200 DNA Analyzer as follows:

$$\%R = \frac{ng/\mu l\ HMW\ DNA\ (>10\ kb)}{ng/\mu l\ total\ DNA} \times 100\%$$

Normalized HMW DNA yield (nY) was calculated as follows using data obtained from the Agilent Technologies TapeStation 2200 DNA Analyzer and extract tissue weights modified by the correction ratio explained above:

$$nY = \frac{\mu g\ HMW\ DNA\ (>10\ kb)}{mg\ extract\ tissue\ weight \times correction\ ratio}$$

Data for each taxon, time interval and response variable (%R and nY) were analyzed separately. A Shapiro-Wilk normality test was used to determine whether data were normally distributed. For each taxon, the effect of storage solution on DNA quality was assessed using a repeated measures design: normally distributed data were analyzed using a parametric repeated measures ANOVA and non-normal data were analyzed using a non-parametric

Friedman $\chi^2$ test (RStudio v. 1.2.1335). Post-hoc tests for repeated measures ANOVAs were performed with the 'nlme' package in RStudio (Version 3.1–137). We used the Friedman Conover test as a post-hoc test for Friedman $\chi^2$ analyses and performed them with a Bonferroni correction in the 'PMCMR' package (Version 4.3).

## Results

DNA was recovered from all samples, taxa, treatments and storage intervals, with the exception of two *A. virens* samples (DESS, 6 months and DE, 6 months), which were lost during DNA extraction. To maintain a fully crossed experiment, all samples derived from these two specimens were excluded from statistical analyses but were included when reporting summary statistics. Total normalized DNA yields ranged from 0.0007 to 12.80 µg DNA/mg tissue and % R ranged from 0.14 to 66.76% across all taxa, treatments and time intervals (S2 Table). Absorbance ratios (A260/A280) can be found in S2 Table.

Qualitative analysis of DNA fragment length distribution, as visualized by agarose gel electrophoresis, showed a similar pattern for all taxa. Specifically, over the duration of the experiment the apparent quantity of observable HMW DNA declined first in tissues stored in DESS-variant solutions that did not contain EDTA (DSS, D and SS). This was evident by one day for tissues of *A. virens* and by 3 months for *M. edulis* and *F. virilis* (S1 Fig). By 6 months, HMW DNA could no longer be visualized in extracts of *M. edulis* and *F. virilis* stored in DESS-variant solutions lacking E (DSS, D and SS) but was evident in all DESS-variant solutions containing E (DESS, ESS and E), as well as in extracts from fresh and EtOH-preserved tissues. For tissues of *A. virens*, by 6 months HMW DNA was no longer observable or appeared only as faint and variable smears in extracts of tissues stored in all DESS-variant solutions (Fig 2).

Quantitative analyses of DNA fragment length distribution were performed using the Agilent Technologies TapeStation 2200 DNA Analyzer. Results for all statistical models appear in Table 2. These data revealed statistically significant differences in %R and/or nY among treatment groups for all taxa and time intervals, even after just one day of storage. For example, after one day of storage in solutions D and DSS, tissues of *M. edulis* showed values of %R significantly lower than those for tissues stored in DESS, DE, ESS and E (Fig 3A). For this same taxon and time interval, nY values for extracts of tissue stored in solution D were also significantly lower than those of DESS, DE, SS and fresh tissue (Fig 4A). In the case of *F. virilis*, the % R value for extracts of tissue stored in solution D for one day was significantly lower than those for DE, E, EtOH and fresh tissue (Fig 3D). No significant differences in nY were observed among treatments for tissues of *F. virilis* at one day (Fig 4D). For *A. virens*, all treatments containing EDTA (DESS, DE, ESS and E) yielded %R values equal to or significantly greater than that recovered from fresh tissue at one day, while those lacking EDTA (DSS, D and SS) yielded values significantly lower than fresh tissue (Fig 3G). For this taxon and time interval, treatment E performed best with respect to nY, giving values statistically indistinguishable from fresh tissue (Fig 4G). In contrast, treatments D and DSS performed most poorly, giving nY values significantly lower than all treatments except SS.

At three months, tissues of *M. edulis* maintained in storage solutions without EDTA (DSS, D and SS) yielded significantly lower %R and nY than fresh tissues or any storage solution containing E (DESS, DE, ESS and E). Similarly, in *F. virilis*, tissues stored in solutions without E (DSS, D and SS) yielded a significantly lower %R and nY than fresh tissue or tissues stored in solutions containing E (DESS, DE, ESS and E), with the exception of DESS, which did not differ significantly from either the best or worst treatments. Additionally, tissues from these two taxa stored in any solution containing E (DESS, DE, ESS and E) or EtOH yielded %R and nY values equal to or significantly greater than fresh tissues at 3 months (Figs 3B, 3E, 4B and 4E).

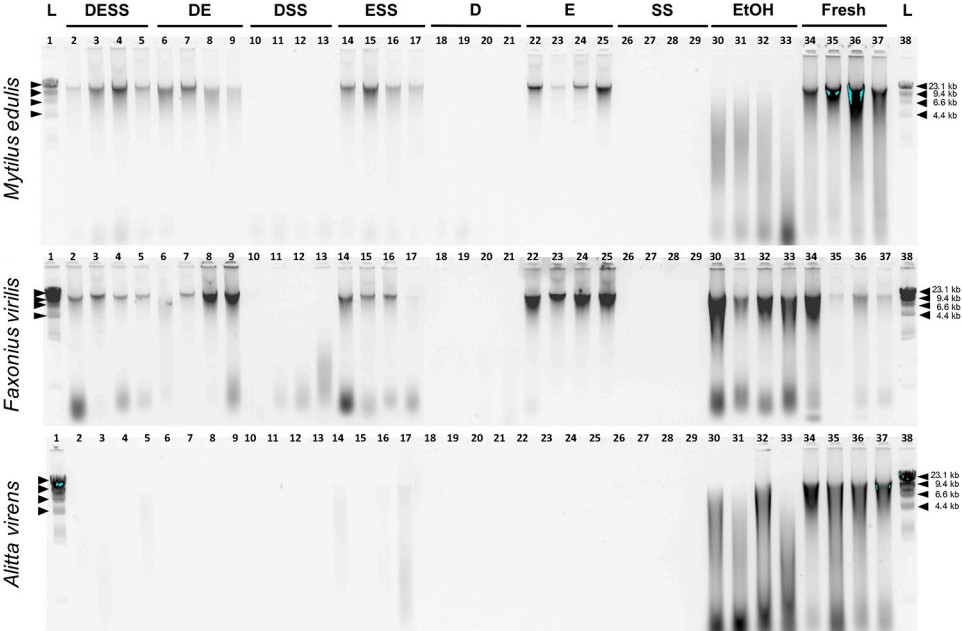

**Fig 2. Qualitative visualization of DNA fragment size distribution after 6 months by agarose gel electrophoresis.** Tissues of three taxa, *Mytilus edulis*, *Faxonius virilis* and *Alitta virens*, were stored for six months at room temperature in DESS (lanes 2–5), six DESS-variant solutions (DE, lanes 6–9; DSS, lanes 10–13; ESS, lanes 14–17; D, lanes 18–21; E, lanes 22–25; SS, lanes 26–29) and 95% ethanol (lanes 30–33). DNA extracts from fresh tissues are displayed in lanes 34–37. Lanes 1 and 38 contain 0.16 µg of λ DNA-HindIII Digest DNA Ladder (New England BioLabs; Ipswich, MA). D, DMSO; E, EDTA; SS, saturated NaCl; EtOH, 95% ethanol; Fresh, untreated tissue extracted immediately after dissection.

In contrast, %R values for all *A. virens* tissue stored in DESS-variant solutions for 3 months were significantly lower than that for fresh tissue (Fig 3H). Nonetheless, *A. virens* tissue stored in DESS, ESS or EtOH were statistically indistinguishable from each other with respect to %R and nY and performed significantly better than tissues preserved in DE, DSS, D and E (Figs 3H and 4H).

After six months of storage, %R and nY values for tissues of both *M. edulis* and *F. virilis* maintained in storage solutions without EDTA (DSS, D and SS) were significantly lower than those for DESS-variants containing EDTA and for those from fresh tissues. Importantly, for both taxa, %R and nY values for tissues maintained in E-containing storage solutions (DESS, DE, ESS and E) were not significantly different from those of fresh tissues or stored in EtOH, with the exception of DE in *M. edulis*, for which %R values were significantly higher than those for EtOH and fresh tissue (Figs 3C, 3F, 4C and 4F). Finally, for *A. virens*, all treatments, except DESS, ESS and EtOH for %R and ESS and EtOH for nY, yielded values that were significantly lower than those for fresh tissue (Figs 3I and 4I).

As an indicator of DNA utility for downstream applications, PCR amplification and sequencing were performed for three randomly selected individuals from fresh tissue of each taxon and tissues stored for six months in treatments DESS or E. With the exception of 1 sample of *A. virens* stored in E, PCR amplifications produced appropriately sized PCR product bands when visualized on agarose gels (S2 Fig). Despite failing to produce a visible PCR band this sample yielded sufficient product for successful sequencing. Of the 27 samples sent for sequencing, 26 samples gave unidirectional sequence and 24 gave bidirectional sequence of at least 500 bp with >94.4% of all base positions yielding quality scores ≥ Q20. The sample that

**Table 2. Summary of statistical model results.**

| | Taxa | Time interval | Statistical test | Test statistic | df | p value |
|---|---|---|---|---|---|---|
| **%R** | *Mytilus edulis* | 1 day | Friedman $\chi^2$ | $\chi^2 = 45.333$ | 8 | < 0.001* |
| | | 3 months | Friedman $\chi^2$ | $\chi^2 = 50.362$ | 8 | < 0.001* |
| | | 6 months | Friedman $\chi^2$ | $\chi^2 = 44.648$ | 8 | < 0.001* |
| | *Faxonius virilis* | 1 day | Repeated measures ANOVA | $F = 2.952$ | 8 | 0.008* |
| | | 3 months | Friedman $\chi^2$ | $\chi^2 = 45.105$ | 8 | < 0.001* |
| | | 6 months | Friedman $\chi^2$ | $\chi^2 = 43.124$ | 8 | < 0.001* |
| | *Alitta virens* | 1 day | Friedman $\chi^2$ | $\chi^2 = 66.080$ | 8 | < 0.001* |
| | | 3 months | Friedman $\chi^2$ | $\chi^2 = 65.387$ | 8 | < 0.001* |
| | | 6 months | Friedman $\chi^2$ | $\chi^2 = 38.067$ | 8 | < 0.001* |
| **nY** | *Mytilus edulis* | 1 day | Friedman $\chi^2$ | $\chi^2 = 24.571$ | 8 | 0.002* |
| | | 3 months | Friedman $\chi^2$ | $\chi^2 = 45.676$ | 8 | < 0.001* |
| | | 6 months | Friedman $\chi^2$ | $\chi^2 = 42.400$ | 8 | < 0.001* |
| | *Faxonius virilis* | 1 day | Friedman $\chi^2$ | $\chi^2 = 11.771$ | 8 | 0.162 |
| | | 3 months | Friedman $\chi^2$ | $\chi^2 = 42.171$ | 8 | < 0.001* |
| | | 6 months | Friedman $\chi^2$ | $\chi^2 = 43.390$ | 8 | < 0.001* |
| | *Alitta virens* | 1 day | Friedman $\chi^2$ | $\chi^2 = 62.400$ | 8 | < 0.001* |
| | | 3 months | Friedman $\chi^2$ | $\chi^2 = 66.693$ | 8 | < 0.001* |
| | | 6 months | Friedman $\chi^2$ | $\chi^2 = 48.100$ | 8 | < 0.001* |

Percent high molecular weight DNA recovered (%R) and normalized high molecular weight DNA yield (nY).

Significant p values ($p < 0.05$) indicated with "*"; df, degrees of freedom.

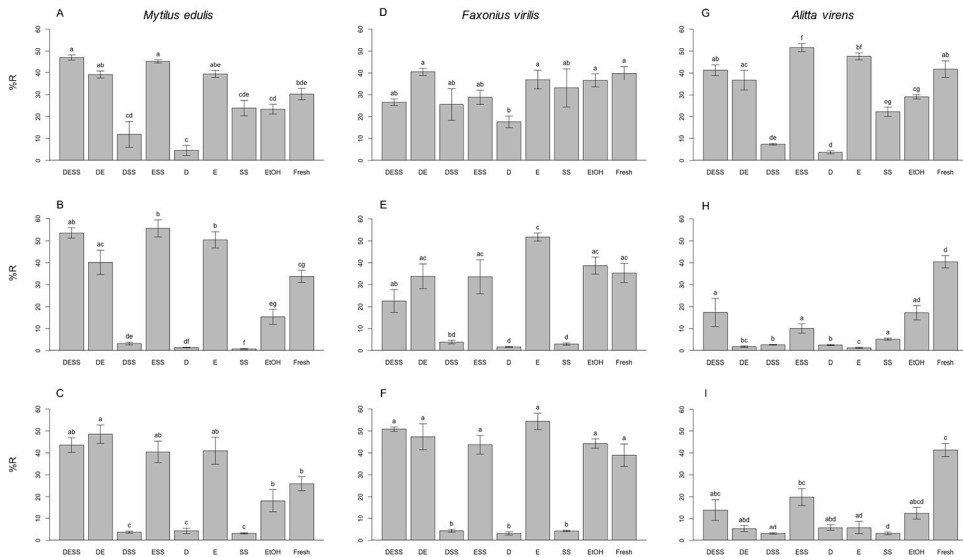

**Fig 3. Percent high molecular weight DNA recovered.** Average percent high molecular weight DNA recovered (%R) was determined for tissues of *Mytilus edulis* (A–C), *Faxonius virilis* (D–F) and *Alitta virens* (G–I) that were extracted immediately from fresh tissue or stored for 1 day, 3 months or 6 months at room temperature in DESS, six DESS-variant solutions or 95% ethanol. Error bars represent standard error. Within each histogram, treatments bearing different lower-case letters are significantly different at $p < 0.05$; matching lower case letters indicate statistically indistinguishable treatments. D, DMSO; E, EDTA; SS, saturated NaCl; EtOH, 95% ethanol; Fresh, untreated tissue extracted immediately after dissection.

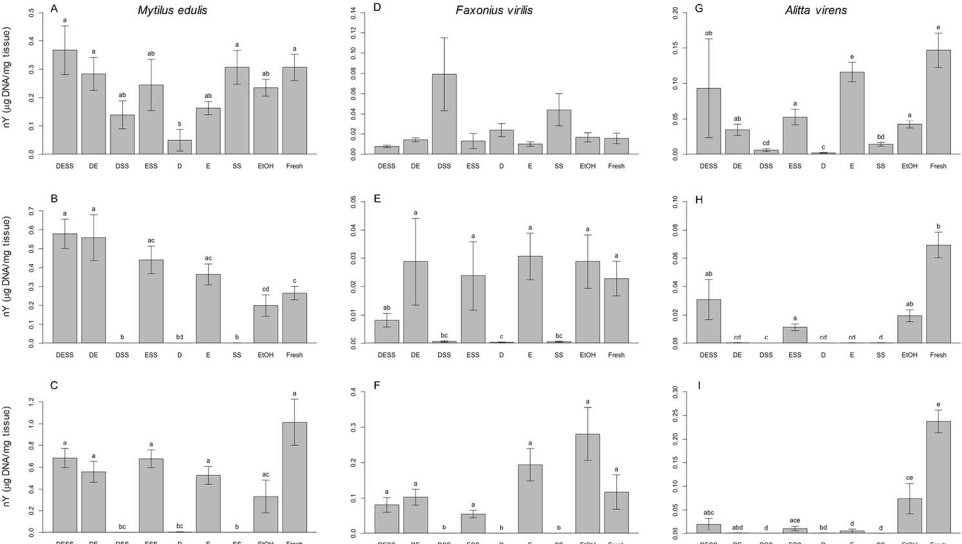

**Fig 4. Normalized high molecular weight DNA yield.** Average normalized high molecular weight DNA yield (nY; μg DNA/mg tissue) was determined for tissues of *Mytilus edulis* (A–C), *Faxonius virilis* (D–F) and *Alitta virens* (G–I) that were extracted immediately from fresh tissue or stored for 1 day, 3 months or 6 months at room temperature in DESS, six DESS-variant solutions or 95% ethanol. Error bars represent standard error. Within each histogram, treatments bearing different lower-case letters are significantly different at $p < 0.05$; matching lower case letters indicate statistically indistinguishable treatments; an absence of letters indicates no significant difference among all treatments in a given model. Note that y-axis scales differ among taxa and time intervals. D, DMSO; E, EDTA; SS, saturated NaCl; EtOH, 95% ethanol; Fresh, untreated tissue extracted immediately after dissection.

failed to produce usable sequence was from fresh tissue of an individual of *M. edulis*. Of the two samples that did not produce bidirectional reads with sufficient quality, one was *F. virilis* preserved in DESS and the other was *A. virens* preserved in E. Although the resulting sequences differed among individual specimens, as should be expected due to within species variation, identical sequences were observed for all PCR products derived from a given individual, regardless of storage treatment. Sequences have been submitted to BOLD and have the following IDs: DESS007-20-DESS032-20.

## Discussion

The quality of DNA obtained from preserved biological specimens can be evaluated in many ways. Here, we chose preservation of high molecular weight (HMW) DNA as a proxy for DNA quality. While we recognize that no single criterion can measure the suitability of a DNA sample for all applications, molecular weight is a simple, useful and easily measurable criterion that provides a first approximation of DNA quality. This is because many forms of DNA damage, including single and double strand breaks, loss or modification of bases and oxidation or chemical modifications of bonds, can directly or indirectly lead to a reduction of average molecular weight.[4, 15, 26, 27] Moreover, HMW DNA is desirable or required for use in many research applications.[2, 28, 29] Indeed, the 'percent above threshold' approach used here has been proposed as a standard metric for reporting DNA quality.[29] Here, we define HMW DNA as DNA with fragment lengths greater than 10 kb. This threshold was selected because this value corresponds roughly with the largest fragment size easily resolvable on agarose gels under typical lab conditions, is comparable to average gene lengths in many higher organisms and is similar to threshold values found in the literature.[11, 28, 29]

We collected our quantitative data using an Agilent Technologies TapeStation 2200 DNA Analyzer and genomic DNA ScreenTapes, which can measure the quantity and size distribution of DNA fragments in a sample over a range from 200 to 60,000 bp in length. We note that by failing to account for the largest and smallest DNA fragments, this method may underestimate %R and nY values for the best-preserved samples and overestimate %R and nY values for the least well-preserved samples. Thus, values at both extremes are expected to be conservative with respect to the model, i.e. less likely to reveal differences among treatments.

Given the finite size of the specimens used, it was not possible to design a factorial experiment that allowed for comparison among all individuals, taxa, treatments and time intervals. Therefore, we chose to limit our statistical analyses to comparing the contributions of each of the three components of DESS to preservation of HMW DNA at a given time interval. We did this by comparing the performance of DESS to solutions containing one of the three components of DESS alone or two components in all pairwise combinations. We chose a factorial design that allowed for statistical comparison of these treatments within a given taxon and time interval. As a result, we do not statistically compare DNA preservation across multiple storage intervals or the effectiveness of individual storage solutions among taxa. This approach allows us to isolate the effect of each component of DESS independent of taxon or specimen-specific effects. While it may be interesting to assess patterns across time and taxa, these additional comparisons would primarily reveal differences in the relative rates of DNA degradation for different taxa rather than giving greater insight into the mechanisms underlying HMW DNA preservation.

In this investigation, several trends were observed in patterns of DNA preservation. Most importantly, DESS-variant solutions containing EDTA performed as well or better than the comparable solution without EDTA. Specifically, for any given taxon and time interval, DESS, DE and ESS yielded equal or significantly greater %R and nY than DSS, D and SS, respectively. Consistent with this observation, solutions without EDTA performed poorly. In fact, we observed less than 5.71%R for all tissues stored in DESS-variant solutions without EDTA (i.e. DSS, D and SS) for all taxa at all time intervals greater than 1 day (Fig 3; S3 Table). This is consistent with a previous study showing that DNA extractions from ant tissue stored in 20% DMSO saturated with NaCl yielded low DNA concentration and poor success in PCR amplification.[30]

By comparison, solutions containing DMSO did not perform better than solutions without DMSO. Specifically, for most taxa and time intervals, solutions containing DMSO did not yield significantly greater %R or nY than those without DMSO (DESS, DE and DSS vs. ESS, E and SS, respectively; Figs 3 and 4). The single exception is that DSS yielded a very small but statistically significant increase in %R as compared to SS for *M. edulis* after storage for 3 months. However, average %R values for both SS (0.75%) and DSS (3.17%) were extremely low as compared to the worst EDTA-containing treatment, DE (40.20%), EtOH (15.37%) or fresh tissue (33.79%) for *M. edulis* at 3 months. Moreover, DSS did not outperform SS with respect nY for this taxon and time interval (S3 Table). Thus, in this investigation, DMSO provided no substantial protection of high molecular weight DNA, nor did it substantially enhance the performance of other components of DESS.

Similarly, saturated NaCl alone provided no significant protection for HMW DNA at time intervals greater than one day. For all taxa, storage in SS resulted in low %R ($\leq$ 3.53%) and nY ($\leq$0.0004 μg DNA/mg tissue). For both *M. edulis* and *F. virilis*, these values were significantly lower than those for fresh tissues or tissues stored in EtOH or any solution containing EDTA. In addition, no significant differences in %R and nY were observed between tissues stored in solutions with or without saturated NaCl (DESS, DSS and ESS vs. DE, D and E, respectively; Figs 3A–3F and 4A–4F). Interestingly, although saturated NaCl alone showed no effect in

preserving HMW DNA, it did appear to provide a slight indirect benefit to the preservation of HMW DNA in certain contexts, i.e. only for *A. virens* and only in the presence of EDTA. At three and six months of storage, the addition of saturated NaCl to storage solutions containing EDTA (i.e. DESS and ESS) slightly but significantly improved %R and nY when compared to tissue stored in solutions containing EDTA without saturated NaCl (i.e. DE and E; Figs 3H, 3I, 4H and 4I). However, when EDTA was not present, the addition of saturated NaCl to another DESS component never significantly improved the %R or nY for any of the tested taxa (i.e. DSS vs. D).

Interestingly, the preservation of HMW DNA in tissues of *A. virens* was poor for all preservatives tested, suggesting differences in the characteristics of the DNAse activity found in its tissue. Most DNase enzymes require magnesium or other divalent cations as cofactors [31, 32] and therefore their activity can be inhibited by divalent cation chelators like EDTA. [33, 34] If the tissue of *A. virens* includes nucleases that are capable of functioning at lower magnesium ion concentrations than those of the other taxa, or if they have greater affinity for magnesium ions than does EDTA, the inhibitory effect of EDTA may be diminished. Consistent with this interpretation, the performance of tested preservative solutions for *A. virens* at one day showed a similar pattern to those observed for the other taxa at 3 and 6 months, suggesting that similar processes may be occurring in all three taxa, although at different rates. The indirect effect of saturated NaCl on preservation by EDTA is also consistent with the potential role of EDTA as a chelator. Salt concentration can alter both the degree of dissociation of EDTA and its ability to chelate divalent cations,[35] potentially changing its effectiveness as a preservative. These hypotheses are testable and will be the topic of future investigations.

Although we show that EDTA provided effective preservation of HMW DNA, evaluating its overall performance as a preservative is beyond the scope of this investigation. Nonetheless, we performed one simple experiment to evaluate the performance of DNA extracts from EDTA-preserved tissues in a common application, PCR amplification and Sanger sequencing. Here, we PCR amplified and sequenced the barcode region[24] of the *COI* gene from DNA extracted from fresh tissues and those preserved in DESS and EDTA for 6 months. We were able to obtain good quality sequence from all samples regardless of preservative treatment, with the exception of one fresh tissue sample of *M. edulis*. Although we observed slightly different sequences among individual specimens, as is expected due to intraspecific variation, all sequences from a given individual were identical regardless of the preservation method.

In conclusion, we found that under conditions in which DESS provided effective preservation of HMW DNA (i.e. resulted in ≥ 20%R), all solutions containing EDTA (DE, ESS and E) were as or more effective than DESS (Fig 3A–3G). This is true for *M. edulis* and *F. virilis* at all time intervals as well as for *A. virens* at one day. Conversely, when DESS was less effective as a preservative (i.e. resulted in < 20%R), none of the six DESS-variant storage solutions provided better protection of HMW DNA than DESS, as seen in *A. virens* after both three and six months of tissue storage (Fig 3H and 3I). These results indicate that for the taxa, treatments and time intervals examined, EDTA is the sole effective preservative component of DESS. These results are surprising in that they indicate that the eponymous ingredients, DMSO and NaCl, may not contribute to the effectiveness of DESS. Furthermore, although EDTA has been used to preserve DNA in blood,[28] it is neither currently in widespread use nor is it widely recognized as a preservative for DNA in other biological tissues. As EDTA is less expensive, easier and safer to make and use than DESS, is not flammable and may be shipped by air without restrictions, continuing research into its efficacy as a tissue preservative is warranted.

## Supporting information

**S1 Table. Species identification.** The barcode region of the mitochondrial *COI* gene was sequenced from two specimens of each taxon used in this study to confirm species identifications. The values listed are percent identities to the best match found in the Barcode of Life Datasystem (BOLD). Specimen IDs for both the Ocean Genome Legacy online catalog and best matches found in BOLD are presented.
(PDF)

**S2 Table. Values for A260/A280 ratios, yield (μg), total normalized DNA yield (μg DNA/mg tissue), normalized high molecular weight DNA yield (nY; μg DNA/mg tissue) and percent high molecular weight DNA recovered (%R) for each sample analyzed in this study.** Values are presented for tissues of *Mytilus edulis*, *Faxonius virilis* and *Alitta virens* extracted immediately after dissection (fresh) or stored for one day (1 d), three months (3 m) or six months (6 m) in preservative treatments containing DMSO (D), EDTA (E) and/or saturated NaCl (SS) or 95% ethanol (EtOH). N/a indicates samples for which data were not collected.
(PDF)

**S3 Table. Average values for yield (μg), total normalized DNA yield (μg DNA/mg tissue), normalized high molecular weight DNA yield (nY; μg DNA/mg tissue) and percent high molecular weight DNA recovered (%R).** Average %R (avg) and standard deviation (SD) for tissues of *Mytilus edulis*, *Faxonius virilis* and *Alitta virens* extracted immediately after dissection (fresh) or stored for one day (1 d), three months (3 m) or six months (6 m) in preservative treatments containing DMSO (D), EDTA (E) and/or saturated NaCl (SS) or 95% ethanol (EtOH).
(PDF)

**S1 Fig. Qualitative visualization of DNA fragment size distribution after 1 day and 3 months by agarose gel electrophoresis.** Tissues of three taxa, *Mytilus edulis*, *Faxonius virilis* and *Alitta virens*, were stored for six months at room temperature in DESS (lanes 2–5), six DESS-variant solutions (DE, lanes 6–9; DSS, lanes 10–13; ESS, lanes 14–17; D, lanes 18–21; E, lanes 22–25; SS, lanes 26–29) and 95% ethanol (lanes 30–33). DNA extracts from fresh tissues are displayed in lanes 34–37. Lanes 1 and 38 contain 0.16 μg of λ DNA-HindIII Digest DNA Ladder (New England BioLabs; Ipswich, MA). D, DMSO; E, EDTA; SS, saturated NaCl; EtOH, 95% ethanol; Fresh, untreated tissue extracted immediately after dissection.
(TIF)

**S2 Fig. Raw agarose gel electrophoresis image for qualitative visualization of *COI* PCR fragment sizes.** Select DNA extracts from all three taxa were PCR amplified after storage for six months. *Mytilus edulis* tissues stored in DESS (lanes 2–4), E (lanes 5–7) or fresh (8–10); *Foxonius virilis* tissues stored in DESS (12–14), E (lanes 15–17) or fresh (18–20); *Alitta virens* tissues stored in DESS (lanes 22–24), E (25–27) or fresh (28–30). Lanes 1, 11 and 21 contain 0.05 μg of Quick-Load Purple 1 kb Plus DNA Ladder (New England BioLabs; Ipswich, MA).
(TIFF)

**S3 Fig. Raw agarose gel electrophoresis images for qualitative visualization of DNA fragment size distribution after 1 day, 3 months and 6 months.** Tissues of *Mytilus edulis*, *Faxonius virilis* and *Alitta virens* were stored for 1 day, three months and six months at room temperature in DESS (lanes 2–5), six DESS-variant solutions (DE, lanes 6–9; DSS, lanes 10–13; ESS, lanes 14–17; D, lanes 18–21; E, lanes 22–25; SS, lanes 26–29) and 95% EtOH (lanes 30–33). DNA extracts from fresh tissues are displayed in lanes 34–37. Lanes 1 and 38 contain 0.16 μg of λ DNA-HindIII Digest DNA Ladder (New England BioLabs; Ipswich, MA). D,

DMSO; E, EDTA; SS, saturated NaCl; EtOH, 95% ethanol; Fresh, untreated tissue extracted immediately after dissection.
(PDF)

**S4 Fig. Tape station outputs.** TapeStation analysis was carried out on DNAs extracted from tissues of *Mytilus edulis*, *Faxonius virilis* and *Alitta virens* that were extracted immediately from fresh tissue or stored for 1 day, 3 months or 6 months at room temperature in DESS, six DESS-variant solutions or 95% ethanol. We provide the gel, sample information and electropherogram including region analysis for each sample analyzed. Values can be cross referenced with S2 Table.
(PDF)

## Acknowledgments

The authors would like to thank the students from Ipswich High School, Gordon College, Endicott College, Northeastern University and the University of Vermont who contributed to this project including Elizabeth Jackson, Kathleen Kelley, Megan Means and Bria Pelletier. The authors would also like to acknowledge the following OGL staff members for their contributions to this project: Rebecca Bernardos, Timery DeBoer, Ann Evankow, Abigail Fusaro and Charlotte Seid.

## Author Contributions

**Conceptualization:** Daniel L. Distel.

**Data curation:** Hannah J. Appiah-Madson, Rosalia Falco.

**Formal analysis:** Hannah J. Appiah-Madson, Rosalia Falco.

**Funding acquisition:** Daniel L. Distel.

**Investigation:** Amy Sharpe, Sonia Barrios, Sarah Gayer.

**Methodology:** Amy Sharpe, Sarah Gayer, Elisha Allan-Perkins, David Stein, Hannah J. Appiah-Madson, Rosalia Falco, Daniel L. Distel.

**Supervision:** Elisha Allan-Perkins, David Stein, Hannah J. Appiah-Madson, Rosalia Falco, Daniel L. Distel.

**Writing – original draft:** Amy Sharpe, Hannah J. Appiah-Madson, Rosalia Falco, Daniel L. Distel.

**Writing – review & editing:** Hannah J. Appiah-Madson, Rosalia Falco, Daniel L. Distel.

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
