## [Decision Letter · Decision Letter 0]

25 May 2020

PONE-D-20-12261

DESS Deconstructed: EDTA as a viable preservative for DNA in biological samples

PLOS ONE

Dear Dr.DISTEL,

Thank you for submitting your manuscript to PLOS ONE. After careful consideration, we feel that it has merit but does not fully meet PLOS ONE’s publication criteria as it currently stands. Therefore, we invite you to submit a revised version of the manuscript that addresses the points raised during the review process. Both reviewers acknowledge the scientific potential of the work. To be accept the paper requires to be implemented with the technical adjustments indicated by both reviewers.

Please submit your revised manuscript by  June 30th, 2020. If you will need more time than this to complete your revisions, please reply to this message or contact the journal office at plosone@plos.org. Please include the following items when submitting your revised manuscript:

We look forward to receiving your revised manuscript.

Kind regards,

Anna Sapino

Academic Editor

PLOS ONE

Journal Requirements:

Reviewers' comments:

Reviewer's Responses to Questions

**Comments to the Author**

1. Is the manuscript technically sound, and do the data support the conclusions?

Reviewer #1: Yes

Reviewer #2: Yes

2. Has the statistical analysis been performed appropriately and rigorously? 

Reviewer #1: Yes

Reviewer #2: Yes

3. Have the authors made all data underlying the findings in their manuscript fully available?

Reviewer #1: No

Reviewer #2: Yes

4. Is the manuscript presented in an intelligible fashion and written in standard English?

Reviewer #1: Yes

Reviewer #2: Yes

5. Review Comments to the Author

Reviewer #1: The study by Sharpe and colleagues evaluated the influence of each component of the DMSO-salt (or DESS) on the preservation of high molecular weight (HMW) DNA purified from 3 aquatic organisms, known to have different fixation success rates. The experimental design is well constructed with at least seven tissue specimens for each organism fixed with DESS or with each DESS component (alone or paired) and analyzed at 3 different time points. Fresh and EtOH-fixed tissues are used as gold-standard specimens. The analyzed variable in the study is the % of DNA with 10kb higher molecular weight. The main results are i) all the fixative combinations comprising the EDTA (E) show a higher ability to preserve HMW DNA ii) DMSO (alone or without E) and SS (NaCl) are the worst fixative compounds iii) the clam worm fixation is the most complex and the less effective.

All experiments and statistics are correctly performed, and the scientific English format is appropriate. However, some points have to be discussed and revised by the authors.

Major points

1. The focus of the research is the qualitative and quantitative DNA assessment after different fixation periods. However, DNA quantification (in the form of total µg of purified DNA) is missing. Moreover, spectrophotometric measurement of the extracted DNA provides other qualitative parameters for the samples (e.g. absorbance ratios). No reports are present in literature about the effect of DESS fixation (and, consequently, of each component) on DNA purity. The authors should add these experiments to improve the novelty of the manuscript.

2. Beside a “fragmentation effect”, other fixative such as formalin can induce DNA sequence alterations, for example C>U deamination, leading to artifactual C>T transition. Do the authors have any evidences about DESS (or each component) effect on DNA biochemical structures?

The authors sequenced the cytochrome c oxidase subunit I (COI) gene for taxa identification. The sequencing of the same gene in a subset of DNA specimens extracted from different organisms, solution components and fixation times can provide information about i) level of amplifiability of the DNA samples ii) presence of sequencing artifacts. Please, perform these additional experiments.

3. In the discussion lines 244-249, the authors described the choices behind the statistical planning, based on the limited cohort size. However, it can be interesting to assess the DNA fragmentation variation within the same type of fixation at the different time points for each independent organism. Please, implement these analyses and discuss them.

4. As discussed in lines 271-274, EDTA alone and EDTA mixed compounds are surprisingly the best fixatives in the study. However, little is known about the EDTA as a single fixative solution. In this context, what are the effect of EDTA fixation from a morphological point of view?

5. The authors obtained the % of HMW DNA from TapeStation capillary electrophoresis. However, none of the agarose gel images or electrophoresis traces obtained from the TapeStation were reported in the manuscript or in the supplementary data. The authors must load these data as supplementary or in a public repository.

Minor points

1. Provide the amount of DNA used in each experiments (DNA amplification, TapeStation and Gel Electrophoresis) in the "Methods" section. Volumes alone are meaningless.

2. As for the sequencing analyses , please report both the concentration and the sequence of the primers used.

3. Add the sequencing instrument used.

4. In Figure 2, clarify the significance of the lower-case letters over the error-bars. These letters are absent in panel F.

5. Add more molecular ladder reference points of the molecular weight in the Figure 3, for increase the robustness of the image.

Reviewer #2: Having worked in the lab where Seutin et al first developed, deployed and published this preservative almost 3 decades ago, and having used it on tens of thousands of samples across dozens of phyla all over the globe - i was most excited to see a breakdown of critical components of this recipe, and i was intrigued by the result. The authors correctly point out that a true multi-factorial investigation into this across time and taxa would be overly onerous, but there a few shortcomings of the experimental design worth discussing, and possibly noting in the manuscript to acknowledge limitations.

First the time frame only extends 6 months. Second, stopping at measuring %HMWDR is not the endpoint of utility of most experiments - we proceed to next steps - and for some to many purposes, a low %HMWDR is not an indicator of experiment success (post-PCR or other genomic applications) - could be acknowledged. Third is the upper bounds and limitations of the measuring DNA quantity and quality using agarose gels and tapestation. For many applications, extremely HMW DNA is valued (de novo genome sequencing on long read platforms), and measuring size on a device like the Femto Pulse or similar would yield much more information - again, not a critical flaw, but could be acknowledged. Next, in the time series where results dipped once (typically A. virens) -they terminated subsequent measurements (keeping the fast/easy agarose gel visualizations) - this presumes that all samples in the series would behave the same and eliminates the possibility of exposing a spurious experimental result in an early time frame that would not persist. just for completeness, the original experimental design should have been completed as designed - we all hate gathering negative data knowingly, but it is still necessary.

With a taxonomic sampling of 3 species (all aquatic invertebrates only - (another limitation that should be acknowledged) - in the end result - what buffer could I or should I use? - 2 of 3 say i can just use "E", but the downside is that 1/3 of the taxa tested actually did better with the original DESS (or ESS) than just E. If I go to the field to preserve a broad diversity of taxa and just take "E", what taxa won't preserve well? If I use DESS, i'll get 3/3, but if i use E, only 2/3. for unknown phyla - we wouldn't necessarily know a priori that E will suffice, and we know that DESS is broadly successful - so why change?

this could all change if the experiments described here (worthy of publication as is) are followed up with next step experimentation - including PCR amplification of small (<1000bp), medium (1000-4000bp) and large (>5000bp) amplicon size fragments and/or genome sequencing on a long read platform to see what library construction effects there might be as a result of the differing preservation regimes.

Why was the data from 3weeks and 6week time interval omitted?

lines 264-265: I'm not sure i concur with the dismissal of a "marginal" improvement in extraction yield because overall extraction was poor for that taxon. in cases where taxa are knowingly going to be difficult - a marginal increase in yield or %HMWDR could be the difference between success and failure at next level experimentation. if adding D or SS are onerous (for all the reasons indicated, and simpler is always better), then it is good to know that just E will suffice most of the time (or for many/most taxa), but not always, and that would be important to know.

The last sentence of the conclusion could be written by a skeptic to add "in 2/3 of aquatic invertebrate phyla". As a user of these preservation buffers, i think my takeaway is a bit more of: in taxa where I am confident there are no preservation issues, I can confidently use E instead of DESS, but if i have no a priori knowledge that E will suffice, then I am better off using DESS to be safe in the knowledge that I'll preserve SOME usable amount of HMWD to then recover it. For this reason, i would like to see a bit of a softening or hedging of the recommendation. When we spend all the time and resources to go collect and preserve biodiversity, we want to ensure those efforts will produce products that will persist.

Finally - this focuses on individual, specimen-based, collecting efforts. Much work is moving into more environmental or mixed/multi-specimen efforts. A similar test conducted on a mixed sample to see what might drop out as recoverable from the mixture would be valuable.

6. PLOS authors have the option to publish the peer review history of their article (what does this mean?). If published, this will include your full peer review and any attached files.

Reviewer #1: No

Reviewer #2: Yes: Lee A Weigt

---

## [Author Response · Author response to Decision Letter 0]

6 Jul 2020

Dear Dr. Sapino,

We thank the reviewers for their insightful and thoughtful comments on the manuscript. In response, we have performed extensive additional analyses and have made major revisions to the manuscript. We feel that the reviewer-suggested modifications have greatly improved the manuscript and hope that you will agree. Because the changes to the manuscript are extensive, we begin our response with a general description of the changes and follow with a point-by-point response to reviewer comments, see below.

Sincerely,

Daniel L. Distel

General: We have revised the manuscript to clarify the intended goal of this investigation. Our aim is to evaluate the contribution of each of the three components of DESS to its ability to preserve high molecular weight DNA in tissues. Based on reviewer comments, we realized that our title and a suggestion made at the end of the discussion may have given the incorrect impression that we also aimed to evaluate the general efficacy of DESS and/or EDTA as preservatives. 

A substantial body of literature already exists to address the efficacy of DESS, as well as the suitability of DNA extracted from DESS-preserved tissues for use in a broad variety of downstream applications. We now review these in lines 61-64. However, little is known about how each of the individual ingredients of DESS contribute to its success. Answering this question is a critical first step toward understanding the mechanisms underlying the preservative activity of DESS and to any future effort to improve its performance. For these reasons, we focused our investigation narrowly on this question.

In general response to reviewer comments, we have: 

1. Changed the title from “DESS deconstructed: EDTA as a viable preservative for DNA in biological samples” to “DESS deconstructed: is EDTA solely responsible for protection of high molecular weight DNA in this common tissue preservative?” This was not made in response to a specific reviewer request, but we felt that it was necessary to clarify the focus of this investigation.

2. Collected, analyzed and reported TapeStation data for all taxa, treatments and time intervals.

3. Added normalized yield of high molecular weight DNA as a second response variable for all taxa, treatments and time intervals.

4. Included all data used in our analyses in supplemental files, including total DNA yield, absorbance ratios at A260/A280 nm, normalized high molecular weight DNA yield, percent high molecular weight DNA recovered and additional images of electrophoresis gels and TapeStation outputs (DESS_S2_Table.pdf, DESS_S4_Fig.pdf). We apologize for the length of the TapeStation data file (S4 Fig; 802 pages). These traces were specifically requested by reviewer 1 and we know of no more compact way to present them.

5. PCR amplified and sequenced DNA extracts for a subset of time intervals and treatments for all taxa.

6. Added more description of quantitative and qualitative results.

7. Discussed what our data may reveal about the mechanism of DNA preservation by DESS.

Specific responses to reviewer comments:

Reviewer 1. 

Comment 1a: The focus of the research is the qualitative and quantitative DNA assessment after different fixation periods. However, DNA quantification (in the form of total µg of purified DNA) is missing. 

Author response: We have now performed and included analyses of normalized high molecular weight DNA yield (abbreviated nY) in addition to our previously reported metric, percent high molecular weight DNA recovered. To accommodate this additional analysis, we have included an additional figure (Figure 4). Note: for clarity we have shortened our abbreviation for percent high molecular weight DNA recovered from %HMWDR to %R.

Comment 1b: Moreover, spectrophotometric measurement of the extracted DNA provides other qualitative parameters for the samples (e.g. absorbance ratios). No reports are present in literature about the effect of DESS fixation (and, consequently, of each component) on DNA purity. The authors should add these experiments to improve the novelty of the manuscript. 

Author response: We now report the total yield of DNA recovered in µg as well as absorbance ratios (A260/A280 nm) for all samples analyzed in this investigation (see S2 Table).

Comment 2: Beside a “fragmentation effect”, other fixative such as formalin can induce DNA sequence alterations, for example C>U deamination, leading to artifactual C>T transition. Do the authors have any evidences about DESS (or each component) effect on DNA biochemical structures? The authors sequenced the cytochrome c oxidase subunit I (COI) gene for taxa identification. The sequencing of the same gene in a subset of DNA specimens extracted from different organisms, solution components and fixation times can provide information about i) level of amplifiability of the DNA samples ii) presence of sequencing artifacts. Please, perform these additional experiments.

Author response: DESS is a preservative rather than a fixative. Unlike formalin, which fixes tissues by chemically reacting with biomolecules, e.g. cross-linking amino groups on DNA and proteins, the components of DESS were selected to preserve and to minimize chemical interaction with DNA. Indeed, DMSO, EDTA and NaCl are widely included in solutions used for DNA storage or to prepare DNA for amplification, cloning, library preparation, sequencing and a host of other applications. For this reason, harmful interactions between DESS components and DNA are not predicted.

Nonetheless, we agree with the reviewer that it would be informative to perform PCR amplification and sequence analysis on a subset of the DNA extracts and have now done so. We have PCR amplified and sequenced the COI gene from a randomly selected subset of DNA extracts from fresh tissue and tissue stored in DESS or EDTA for six months for each of the three taxa in used this study. Our results show that storage in DESS or EDTA for six months did not prevent successful PCR amplification and accurate DNA sequencing of this marker gene (see lines 188-190, 282-292 and 366-373 and S2 Fig).

Comment 3: In the discussion lines 244-249, the authors described the choices behind the statistical planning, based on the limited cohort size. However, it can be interesting to assess the DNA fragmentation variation within the same type of fixation at the different time points for each independent organism. Please, implement these analyses and discuss them.

Author response: Although we agree that these analyses would be interesting, we argue that they do not address the questions that we pose in this manuscript. Analyzing differences between time points and between taxa would provide information about species-specific rates of DNA degradation but would not provide information about the relative contributions of DESS components to preservation of high molecular weight DNA. In other words, these results would be applicable to the particular species and tissues examined but would not provide any generalizable information about mechanism. Moreover, to perform statistical analyses to evaluate the rates of DNA fragmentation over time and between taxa would require an entirely different experimental design and a new study of approximately twice the size, duration and complexity of the study already completed. We hope that the reviewer will agree that these analyses are beyond the scope of this investigation and, although interesting and suitable for future study, would not constitute an effective use of time and resources in the context of the present research question.

Comment 4: As discussed in lines 271-274, EDTA alone and EDTA mixed compounds are surprisingly the best fixatives in the study. However, little is known about the EDTA as a single fixative solution. In this context, what are the effect of EDTA fixation from a morphological point of view?

Author response: DESS is an effective preservative but is not an effective fixative. It contains two excellent solvents, water and DMSO. The latter is not only an excellent solvent for both polar and nonpolar compounds but also mobilizes many solutes across cell membranes. In addition, EDTA is a powerful divalent cation chelator which causes rapid dissolution of calcareous structures including bones, teeth, shells and tests. Thus, DESS and its components are not suitable for fixation of morphology or ultrastructure, and with few exceptions are not used for this purpose. 

Comment 5: The authors obtained the % of HMW DNA from TapeStation capillary electrophoresis. However, none of the agarose gel images or electrophoresis traces obtained from the TapeStation were reported in the manuscript or in the supplementary data. The authors must load these data as supplementary or in a public repository.

Author response: We apologize for this oversight. We have now included this information as supplementary data (see S4 Fig).

Minor points

Comment 1: Provide the amount of DNA used in each experiments (DNA amplification, TapeStation and Gel Electrophoresis) in the "Methods" section. Volumes alone are meaningless.

Author response: We now report concentration in addition to volume for template DNAs used in the PCR amplifications. However, volume is the appropriate metric for the TapeStation and electrophoresis experiments. This is because the TapeStation and gel electrophoresis were the methods used for quantifying and visualizing DNA concentration. Therefore, by design, the concentrations of DNA in the samples loaded on tapes and gels were not known before they were determined using these methods. Nonetheless, we now provide the averages and a table listing the concentrations and amounts of DNA determined for each sample using these methods (see lines 176-177, 181 and 184 and S2 Table, respectively). 

Comment 2: As for the sequencing analyses, please report both the concentration and the sequence of the primers used.

Author response: We have now done so. Please see lines 127-130.

Comment 3: Add the sequencing instrument used.

Author response: This has now been added. Please see line 135.

Comment 4: In Figure 2, clarify the significance of the lower-case letters over the error-bars. These letters are absent in panel F.

Author response: We have now provided a more explicit description of their meaning on lines 251-252 and 258-260. Please note that the absence of lower-case letters in a histogram indicates no significant differences among all treatments for that statistical model. This is now explained in line 260.

Comment 5: Add more molecular ladder reference points of the molecular weight in the Figure 3, for increase the robustness of the image.

Author response: We have now done so.

Reviewer #2: 

Having worked in the lab where Seutin et al first developed, deployed and published this preservative almost 3 decades ago, and having used it on tens of thousands of samples across dozens of phyla all over the globe - i was most excited to see a breakdown of critical components of this recipe, and i was intrigued by the result. The authors correctly point out that a true multi-factorial investigation into this across time and taxa would be overly onerous, but there a few shortcomings of the experimental design worth discussing, and possibly noting in the manuscript to acknowledge limitations.

Comment 1: First the time frame only extends 6 months. 

Author response: We agree with the reviewer that longer time intervals are always preferable in preservative experiments. To provide better context for the intervals chosen, we surveyed 14 publications in which time intervals of preservation were tested for DESS and found that six months was the median preservation interval tested. This indicates that this interval is consistent with the norm for publication and provides a serviceable window for many applications where tissue must be preserved before analyses. We now report this on lines 67-69. 

More importantly, we note that that the aim of our investigation is not to evaluate the suitability of DESS as a preservative. There is an abundant body of literature addressing this question (reviewed in lines 64-68). Instead our aim was to evaluate the contribution of the various components of DESS to its preservative activity. We found that by six months, the normalized yield of HMW DNA in any DESS variant that lacked EDTA was close to our lowest limit of detection, indicating near complete degradation (see Fig 4). Therefore, we conclude that longer time points would not strengthen our central finding. 

Comment 2: Second, stopping at measuring %HMWDR is not the endpoint of utility of most experiments - we proceed to next steps - and for some to many purposes, a low %HMWDR is not an indicator of experiment success (post-PCR or other genomic applications) - could be acknowledged. 

Author response: We now acknowledge that no single metric, including high molecular weight, can adequately measure the suitability of a given DNA extract for all downstream applications. Please see lines 296-298.

Comment 3: Third is the upper bounds and limitations of the measuring DNA quantity and quality using agarose gels and tapestation. For many applications, extremely HMW DNA is valued (de novo genome sequencing on long read platforms), and measuring size on a device like the Femto Pulse or similar would yield much more information - again, not a critical flaw, but could be acknowledged. 

Author response: We have done so. Please see lines 307-312. 

Comment 4: Next, in the time series where results dipped once (typically A. virens) -they terminated subsequent measurements (keeping the fast/easy agarose gel visualizations) - this presumes that all samples in the series would behave the same and eliminates the possibility of exposing a spurious experimental result in an early time frame that would not persist. just for completeness, the original experimental design should have been completed as designed - we all hate gathering negative data knowingly, but it is still necessary.

Author response: We have performed the requested TapeStation analyses on all samples and have provided the data in the revised draft and in Figs 3 and 4.

Comment 5: With a taxonomic sampling of 3 species (all aquatic invertebrates only - (another limitation that should be acknowledged) - in the end result - what buffer could I or should I use? - 2 of 3 say i can just use "E", but the downside is that 1/3 of the taxa tested actually did better with the original DESS (or ESS) than just E. If I go to the field to preserve a broad diversity of taxa and just take "E", what taxa won't preserve well? If I use DESS, i'll get 3/3, but if i use E, only 2/3. for unknown phyla - we wouldn't necessarily know a priori that E will suffice, and we know that DESS is broadly successful - so why change? this could all change if the experiments described here (worthy of publication as is) are followed up with next step experimentation - including PCR amplification of small (<1000bp), medium (1000-4000bp) and large (>5000bp) amplicon size fragments and/or genome sequencing on a long read platform to see what library construction effects there might be as a result of the differing preservation regimes.

Author response: This is an excellent point. In retrospect we realize that suggesting that EDTA may be a suitable substitute for DESS goes beyond the intended aims of our manuscript. This statement has therefore been removed from the manuscript. 

Comment 6: Why was the data from 3weeks and 6week time interval omitted?

Author response: We examined five time intervals qualitatively using gel electrophoresis to establish an appropriate time course for the investigation. Our aim was to ensure that we used storage intervals long enough to reveal measurable differences in DNA preservation, but not so long as to result in complete degradation within all treatments for any individual taxon. Using qualitative methods, we established that observable differences in preservation of high molecular weight DNA could already be detected at the first time point (one day), and that high molecular weight DNA could still be detected in at least some treatments for all taxa at the longest time point (six months). This showed us that one day to six months was a suitable time frame and allowed us to select three time points within that time frame, the shortest interval, the midpoint and the longest interval for quantitative analyses. We now explain this in lines 146-152. 

Comment 7: lines 264-265: I'm not sure i concur with the dismissal of a "marginal" improvement in extraction yield because overall extraction was poor for that taxon. in cases where taxa are knowingly going to be difficult - a marginal increase in yield or %HMWDR could be the difference between success and failure at next level experimentation. if adding D or SS are onerous (for all the reasons indicated, and simpler is always better), then it is good to know that just E will suffice most of the time (or for many/most taxa), but not always, and that would be important to know. The last sentence of the conclusion could be written by a skeptic to add "in 2/3 of aquatic invertebrate phyla". As a user of these preservation buffers, i think my takeaway is a bit more of: in taxa where I am confident there are no preservation issues, I can confidently use E instead of DESS, but if i have no a priori knowledge that E will suffice, then I am better off using DESS to be safe in the knowledge that I'll preserve SOME usable amount of HMWD to then recover it. For this reason, i would like to see a bit of a softening or hedging of the recommendation. When we spend all the time and resources to go collect and preserve biodiversity, we want to ensure those efforts will produce products that will persist.

Author response: This is a point well taken. We have eliminated this recommendation.

Comment 8: Finally - this focuses on individual, specimen-based, collecting efforts. Much work is moving into more environmental or mixed/multi-specimen efforts. A similar test conducted on a mixed sample to see what might drop out as recoverable from the mixture would be valuable.

Author response: This is an excellent suggestion that we will keep in mind for future investigations.

---

## [Decision Letter · Decision Letter 1]

24 Jul 2020

DESS deconstructed: is EDTA solely responsible for protection of high molecular weight DNA in this common tissue preservative?

PONE-D-20-12261R1

Dear Dr. Distel,

We’re pleased to inform you that your manuscript has been judged scientifically suitable for publication and will be formally accepted for publication once it meets all outstanding technical requirements.

Kind regards,

Anna Sapino

Academic Editor

PLOS ONE

Additional Editor Comments (optional):

Reviewers' comments:

Reviewer's Responses to Questions

**Comments to the Author**

1. If the authors have adequately addressed your comments raised in a previous round of review and you feel that this manuscript is now acceptable for publication, you may indicate that here to bypass the “Comments to the Author” section, enter your conflict of interest statement in the “Confidential to Editor” section, and submit your "Accept" recommendation.

Reviewer #1: All comments have been addressed

Reviewer #2: All comments have been addressed

2. Is the manuscript technically sound, and do the data support the conclusions?

Reviewer #1: Yes

Reviewer #2: Yes

3. Has the statistical analysis been performed appropriately and rigorously? 

Reviewer #1: Yes

Reviewer #2: Yes

4. Have the authors made all data underlying the findings in their manuscript fully available?

Reviewer #1: Yes

Reviewer #2: Yes

5. Is the manuscript presented in an intelligible fashion and written in standard English?

Reviewer #1: Yes

Reviewer #2: Yes

6. Review Comments to the Author

Reviewer #1: The authors replied to all the questions in a very efficient manner. In particular, the introductory summary concerning the manuscript modifications is surrounded by clear justifications and allows to understand all the subsequent confutations. As for the point 2, the authors commented that DESS is a preservative rather than a fixative. I agree, but the same authors defined DESS as a "fixative" in lane 64, page 3 in the first manuscript. They modified in this second turn of revision the term "fixative “with "preservative", to avoid misleading significance. I also agree to the reply of point 3, and I also suggest to design further studies to understand the effect on DNA fragmentation caused by different preservation times, solutions and taxa. As for the S4 figure size, supplementary data are unlimited, and in this case they can be useful as examples of TapeStation traces for other researchers focused on the study of tissue preservation / fixation. The manuscript is now more comprehensive, no other revisions are needed.

Reviewer #2: The revised manuscript is a significant improvement and has addressed all concerns raised in the previous review.

7. PLOS authors have the option to publish the peer review history of their article (what does this mean?). If published, this will include your full peer review and any attached files.

Reviewer #1: No

Reviewer #2: **Yes: **Lee A Weigt

---

## [Editor Report · Acceptance letter]

30 Jul 2020

PONE-D-20-12261R1 

DESS deconstructed: is EDTA solely responsible for protection of high molecular weight DNA in this common tissue preservative? 

Dear Dr. Distel:

I'm pleased to inform you that your manuscript has been deemed suitable for publication in PLOS ONE. Congratulations! Your manuscript is now with our production department. 

Kind regards, 

on behalf of

Dr. Anna Sapino 

Academic Editor

PLOS ONE